



**Effects of grass leaf anatomy, development and light/dark alternation on the triple oxygen isotope signature of leaf water and phytoliths: insights for a new proxy of continental atmospheric humidity**

Anne Alexandre[1], Elizabeth Webb[2], Amaelle Landais[3], Clément Piel[4], Sébastien Devidal[4], Corinne Sonzogni[1], Martine Couapel[1], Jean-Charles Mazur[1], Monique Pierre[2], Frédéric Prié[2], Christine Vallet-Coulomb[1], Jacques Roy[4].

[1]Aix Marseille Univ, CNRS, IRD, INRA, Coll France, CEREGE, Aix-en-Provence, France

[2]Department of Earth Sciences, The University of Western Ontario, London, Ontario, Canada

[3]Laboratoire des Sciences du Climat et de l'Environnement (LSCE/IPSL/CEA/CNRS/UVSQ), Gif-sur-Yvette, France

[4]Ecotron Européen de Montpellier, UPS 3248, Centre National de la Recherche Scientifique (CNRS), Campus Baillarguet, Montferrier-sur-Lez, France

Correspondance: alexandre@cerege.fr

**Abstract**

Continental relative humidity (RH) is a key-climate parameter. However, there is a lack of quantitative RH proxies suitable for climate model-data comparisons. Recently, a combination of climate chamber and natural transect calibrations laid the groundwork for examining the robustness of the triple oxygen isotope composition ($\delta^{18}O$, $\delta^{17}O$) of phytoliths as a new proxy for past changes in RH. However, it was recommended that besides RH, additional factors that may impact $\delta^{18}O$ and $\delta^{17}O$ of plant water and phytoliths be examined. Here, the effects of leaf anatomy, leaf development stage and day/night alternations are addressed from the growth of the grass species *F. arundinacea* in climate chambers. Plant water and phytoliths are analyzed in $\delta^{18}O$ and $\delta^{17}O$. Silicification patterns are examined using light and scanning electron observation of phytoliths. The isotope data show the increasing contribution of evaporated epidermal water to the bulk leaf water, from sheath to proximal and apical leaf blade. However, despite this isotope heterogeneity, $\delta^{18}O$ and $\delta^{17}O$ of the bulk leaf water can be predicted by the Craig and Gordon model, in the given experimental conditions (high RH). Regarding phytoliths, their forming water (mainly epidermal) is, as expected, more impacted by evaporation than the bulk leaf water. This discrepancy increases from sheath to proximal and apical blade and can be explained by the steepening of the radial concentration gradient of evaporated water along the leaf. However, we show that because most of silica polymerizes in epidermal long cells of the apical blade of the leaves, the $\delta^{18}O$ and $\delta^{17}O$ of bulk grass phytoliths should not be impacted by the diversity in grass anatomy. The data additionally show that most of silica polymerizes at the end of the leaf elongation stage and at the transition towards leaf senescence. Thus, climate conditions at that time should be considered when interpreting $\delta^{18}O$ and $\delta^{17}O$ of phytoliths from the natural environment. At least, no light/dark effect was detected on the $\delta^{18}O$ and $\delta^{17}O$ signature of plant water and phytoliths of *F. arundinacea*. However, when day/night alternations are characterized by significant changes in RH, the lowest RH conditions favoring evaporation and silica polymerization should be considered when calibrating the phytolith proxy. This study contributes to the identification of the parameters driving the $\delta^{18}O$ and $\delta^{17}O$ of bulk grass phytoliths. It additionally brings elements to further understand and model the $\delta^{18}O$ and $\delta^{17}O$ of grass leaf water, which influences the isotope signal of several processes at the soil/plant/atmosphere interface.

## 1. Introduction

Recently, a combination of growth chamber and natural transect calibrations laid the groundwork for examining the robustness of the triple oxygen isotope composition of phytoliths as a new proxy for past changes in continental atmospheric relative humidity (RH) (Alexandre et al., 2018). Continental RH is a key-





climate parameter. When combined with atmospheric temperature, it can be used to estimate the concentration of atmospheric water vapor, one of the main components of the global water cycle. However, global climate models have difficulties to properly capture continental RH (Sherwood et al., 2010; Risi et al., 2012; Fischer and Knutti, 2013) and there is a lack of RH proxies suitable for model-data comparisons.

Phytoliths are micrometric particles of hydrous amorphous silica ($SiO_2 (H_2O)_n$) that form in and between living
plant cells. Silica polymerizes from the sap that contains dissolved silicon (among other nutrients) absorbed by the plant roots from the soil. Phytoliths, can take the shape of the cells they form in, which gives them morphological taxonomic properties. Hence, phytolith morphological assemblages extracted from buried soils, loess and sediments are commonly used for paleoenvironmental reconstructions (e.g. Miyabuchi and Sugiyama, 2015; Nogué et al., 2017; Woodburn et al., 2017). In grasses, silica that can represents several
percent of the dry weight (d.w.) (Alexandre et al., 2011), polymerizes mainly in the stem and leaf epidermis from where the plant water evaporate during transpiration (Kumar et al., 2016). This polymerization is assumed to occur in isotope equilibrium with the plant water (Alexandre et al., 2018; Shahack-Gross et al., 1996; Webb and Longstaffe, 2000).

Variations in the triple oxygen isotope composition ($\delta^{18}O$, $\delta^{17}O$) of natural surficial waters, and consequently
of minerals formed in isotope equilibrium with these waters (Gázquez et al., 2015; Herwartz et al., 2017; Passey et al., 2014) are mainly driven by the extent of kinetic fractionation during evaporation. To the contrary, they are only weakly sensitive to distillation processes (Angert et al., 2004; Barkan and Luz, 2007; Landais et al., 2008; Uemura et al., 2010; Steig et al., 2014) and not significantly affected by temperature (Barkan and Luz, 2005; Uemura et al., 2010), in contrast to variations in the deuterium-excess (d-excess = $\delta^2H - 8.0$ x
$\delta^{18}O$). This makes the triple oxygen isotope composition of surface waters a powerful tool for tracing evaporative conditions (Luz and Barkan, 2010; Surma et al., 2018, 2015) that are mainly dependent on atmospheric RH (Cernusak et al., 2016; Craig and Gordon, 1965). In plant water, changes in the triple oxygen isotope composition due to evaporation during transpiration, can exceed most of the variations identified so far in seawater, surface water, rainfall and ice (Landais et al., 2006; Li et al., 2017; Sharp et al., 2018)

In Alexandre et al. (2018) the triple oxygen isotope composition of phytoliths from a grass species (*Festuca arundinacea*) grown under controlled conditions of RH, the other climate parameters being set constant, was examined. A linear relationship with RH was demonstrated. The triple oxygen isotope composition of soil phytoliths collected in West Africa along a vegetation and RH transect where soil water availability and climate parameters other than RH changed simultaneously, also showed a linear regression with RH, close to
the correlation obtained from the growth chamber. This linear regression allowed for the prediction of RH with a standard error of 5.6%. However, it was recommended that besides RH, additional factors that may impact the isotope composition of bulk phytolith samples be examined, before using this relationship for paleoenvironmental reconstruction.

In particular, in nature, the biomass of grass stem, sheath and blade, as well as the geometry of leaf blade is
highly species-dependant. Previous studies showed that for grasses the water $\delta^{18}O$ increases from stem to blade and from the bottom to the tip of the blade (e.g. Helliker and Ehleringer, 2000; Farquhar and Gan, 2003; Webb and Longstaffe, 2003; Cernusak et al., 2016). Meanwhile, the $^{17}O$-excess decreases (Landais et al., 2006). This raises the question of the effect of grass anatomy diversity on the triple oxygen isotope signal of bulk phytolith samples. Moreover, climate parameters (e.g. mean annual RH or mean RH of the growing season; mean daily
RH or RH at 12:00 UTC…, etc) to be considered when interpreting the triple oxygen isotope signature of bulk grass phytoliths recovered from soils or sediments depends on when most of the silica deposition occurs in plants. In grasses, silica deposition can be either metabolically controlled or passive, i.e. depending mainly on silica saturation during cell dehydration when the leaf water evaporates (Kumar et al., 2017 and references therein, Kumar et al., 2019). The contribution of evaporated water to the bulk leaf water is expected to vary
with transpiration (Cernusak et al., 2016) which decreases from day to night (Caird et al., 2007) and when senescence occurs (Norton et al., 2014). This raises the question of the impact of senescence and day/night alternations on the triple oxygen isotope composition of bulk leaf water and phytoliths.





In order to address these issues, a first growth chamber experiment was set up to explore the spatial heterogeneity in $\delta^{18}O$ and $\delta^{17}O$ of water and phytoliths in *F. arundinacea*. Bulk water and phytoliths were extracted from sheaths, stems, proximal and apical leaves, young, mature and senescent bulk leaves. Silica concentration was measured and phytolith morphological assemblages, that give information on the type of tissue and cells that are silicified, were examined in the same samples. A second experiment was set up to explore light/dark and day/night alternations on $\delta^{18}O$ and $\delta^{17}O$ of plant water and phytoliths of *F. arundinacea*. Implications for the calibration of the triple oxygen isotope composition of phytoliths as a RH proxy are discussed in the light of the newly acquired results.

## 2. The triple oxygen isotope composition of waters and minerals: notations

In the oxygen triple isotope system ($\delta^{18}O$, $\delta^{17}O$), the fractionation factors ($^{17}\alpha$ and $^{18}\alpha$) are related by the exponent $\theta$ where $^{17}\alpha = {}^{18}\alpha^{\theta}$ or $\theta = \ln^{17}\alpha / \ln^{18}\alpha$. $\theta$ can also be formulated as $\theta = \Delta'^{17}O_{A-B} / \Delta'^{18}O_{A-B}$ with $\Delta'^{17}O_{A-B} = \delta'^{17}O_A - \delta'^{17}O_B$, $\Delta'^{18}O_{A-B} = \delta'^{18}O_A - \delta'^{18}O_B$, $\delta'^{17}O = \ln(\delta^{17}O + 1)$ and $\delta'^{18}O = \ln(\delta^{18}O + 1)$. $\delta$ and $\delta'$ notations are expressed in ‰ vs VSMOW. For atmospheric temperature conditions, when isotope equilibrium is reached between water and silica or between liquid and vapor, $\theta$ stays quasi-constant. For water-silica, $\theta_{silica-water}$ equals 0. 524 for the 5-35°C range (Sharp et al., 2016). For water liquid-water vapour $\theta_{equil}$ equals 0.529 for the 11-41°C range (Barkan and Luz, 2005). When evaporation occurs, a fractionation due to the vapour diffusion in air is added to the equilibrium fractionation, as conceptualized by the Craig and Gordon model (Craig and Gordon, 1965; Gat, 1996). $\theta_{diff}$ associated with this diffusion fractionation equals 0.518 (Barkan and Luz, 2007). When RH decreases, amplitude of the diffusion fractionation governed by $\theta_{diff}$ increases. $\theta$ is linked to a particular physical process. When several processes occur at the same time, the resulting fractionation exponent is rather named $\lambda$. In the $\delta'^{18}O$ vs $\delta'^{17}O$ space, $\lambda$ represents the slope of the line linking $\Delta'^{17}O_{A-B}$ to $\Delta'^{18}O_{A-B}$. During evaporation $\lambda$ should basically decrease from 0.529 ($\theta_{equil}$) to 0.518 ($\theta_{diff}$).

Another term used in hydrological studies is $^{17}O$-excess ($^{17}O$-excess = $\delta'^{17}O - 0.528 \times \delta'^{18}O$, Luz and Barkan, 2010) which, in the $\delta'^{17}O$ vs $\delta'^{18}O$ space, is the departure from a reference line with the slope $\lambda$ of 0.528. This reference line is the trend along which meteoric waters were initially shown to plot (Luz and Barkan, 2010). As it is close to the liquid-vapor equilibrium exponent $\theta_{equil}$ (0.529), its use makes the $^{17}O$-excess very convenient to highlight kinetic processes that result from evaporation. As the magnitudes of changes of the triple oxygen isotope composition in the water cycle and terrestrial minerals are very small, the $^{17}O$-excess is expressed in per meg (per meg = $10^{-3}$‰). Here, we additionally use the difference in $^{17}O$-excess$_{A-B}$ associated with a chain of processes from A to B ($^{17}O$-excess$_{A-B} = \Delta'^{17}O_{A-B} - 0.528 \times \Delta'^{18}O_{A-B}$) (Hayles et al., 2018).

## 3. Material

In three growth chambers, the grass species *F. arundinacea* was sown and grown in commercial potting soil in a 35 L container (53 x 35 x 22 cm LxWxD). Ten days after germination, agar-agar was spread on the soil surface around the seedlings, to prevent any evaporation from the soil as described in Alexandre et al. (2018). Ambient RH was kept constant in the growth chamber by combining a flow of dry air and an ultrasonic humidifier that produces vapour without any isotope fractionation. The vapour and the soil irrigation water (IW) came from the same source and their triple oxygen isotope composition was similar (-5.6 ± 0.0 ‰ and 26 ± 5 per meg for $\delta'^{18}O$ and $^{17}O$-excess, respectively).

**Experiment 1.** This experiment was designed to examine the grass leaf water and phytolith isotope signatures in different parts of the leaf and at different stages of the leaf development. Briefly, the stages taken into account were i) young leaf, where only the end of the blade is visible as it emerges from the sheath of the preceding leaf, ii) adult leaf where the blade is fully developed, the ligule visible and the sheath is well formed, and iii) yellow and desiccated senescent leaf.

*F. arundinacea* was grown for 39 days, in a climate chamber where light, air temperature and RH were set constant at 290 µmol/m²/sec, 20°C and 73% respectively. On day 28, irrigation was stopped to force



senescence of the leaves. Eleven days later, a total of 197g of biomass was collected. From this biomass, young
leaf (visible end of the blade), adult leaf and senescent leaf (blade only) were separated. Adult leaves were
sectioned into three parts: sheath, proximal part of the blade (10 cm long) and apical part of the blade. Five
samples resulted (Table 1). For all samples except the senescent leaves, three to five g of biomass were put in
gastight glass vials and kept frozen for bulk leaf water extraction. Senescent leaves were too dry for water
extraction. The rest of the biomass (between 10 and 70 g depending on the sample, (Table 1) was dried for
phytolith extraction.

**Experiment 2.a** Light triggers the opening of plant stomata with, as an inevitable consequence, an increase in
water loss through these stomata. At night, however, stomata often do not close totally. Night transpiration is
often 5 to 15 % of day transpiration (Caird et al., 2007). In *F. arundinacea*, stomatal conductance at night can
be as high as 30 % of conductance during the day (Pitcairn et al., 1986). Together with difference in air RH
between day and night, this could affect isotope enrichment of leaves (Barbour et al., 2005). This experiment
was thus designed to assess whether light/dark alternation may impact the isotope signature of *F. arundinacea*
leaf water.

In a growth chamber, *F. arundinacea* was grown for 22 days with constant light. Then, a 12h light/12h dark
alternation was introduced. Temperature and RH were kept constant at 25°C and 60% respectively. Half of
the biomass was harvested at the end of day 22 (constant light). The second half was harvested at the end of
the alternation period on day 26. In order to consider potential spatial heterogeneity, leaf blades (both young
and adult leaves) were collected from four different places in the culture for each harvest. The eight resulting
samples (Table 2) were put in gastight glass vials and kept frozen for bulk leaf blade water extraction.

**Experiment 2.b** In natural conditions, day/night alternations imply changes in temperature and RH in addition
to changes in light intensity. This experiment was designed to assess whether over a period of several day/night
alternations, changes in RH during the night impacted the mean isotope signature of grass leaf phytoliths.

For this experiment, the leaf water was not analyzed as it only gives a snapshot of its isotope composition. *F.
arundinacea* was grown in two growth chambers. In the first chamber, light, temperature and RH were kept
constant (290 mmol/m2/sec, 25°C and 60%, respectively). In the second chamber, 12H day/12H night
alternations were set. During the day (light 290 μmol/m2/sec), temperature and RH were set to 25°C and 60%,
respectively, whereas during the night (no light), they were set to 20°C and 80%, respectively. The leaf blades
(both young and adult leaves) were harvested after a first growth of 16 days and a second growth of 18 days
(Table 2) and dried for phytoliths extraction.

## 4. Methods

### 4.1. Phytolith chemical extraction, counting and analysis

Phytoliths were extracted using a high purity protocol with HCl, $H_2SO_4$, $H_2O_2$, $HNO_3$, $KClO_3$ and KOH at 70
°C following Corbineau et al. (2013) and Alexandre et al. (2018). Phytoliths were weighed and their mass
reported to the initial leaf dry weight (d.w.). To account for leaf mass loss during senescence, a mass loss
correction factor of 0.7, previously estimated for graminoids (Vergutz et al., 2012) was applied to the phytolith
concentration in senescent leaves (Table 1a).

Most grass phytoliths have a morphology characteristic of their cell of origin. Phytolith morphological
assemblages were thus determined to follow the spatial evolution over time of the leaf silicification. Phytoliths
assemblages from experiment 1 were mounted on microscope slides in Canada Balsam, and counted in light
microscopy at a 600X magnification. More than 200 phytoliths with a dimension greater than 5 μm and with a
characteristic morphology were counted. Phytolith types were named using the International Code for Phytolith
Nomenclature 1.0 (Madella et al., 2005) and categorized as follows: Trapeziform short cell and Trapeziform
sinuate short cell coming from the short cell silicification, Elongate cylindric and Elongate echinate coming
from the intercoastal long cell silicification, Acicular produced by hair silicification and Parallelepipedal
produced by bulliform cells silicification (Table 1, fig. 1). These characteristic phytoliths are commonly used
for paleoenvironmental reconstructions when recovered from buried soils or sediments (e.g. Woodburn et al.,



2017). In addition, thin silica particles with uncharacteristic shape and with a refractive index too low to be accurately described using light microscopy were also counted. Abundance of the phytolith categories are expressed in % of the sum of counted particles. Three repeated counts usually give an error lower than ± 5 % (SD).

The phytolith assemblages were further observed with a Scanning Electron Microscope (FEG-SEM, HITACHI SV6600, accelerating voltage of 3KV, 15degree tilt, working distance of 14mm and probe current of a few pA to avoid charging issues), after carbon coating.

Phytoliths triple oxygen isotope analysis was performed as described in details in Alexandre et al. (2018). The IR Laser-Heating Fluorination Technique (Alexandre et al., 2006, Crespin et al., 2008; Suavet et al., 2010)
was used to extract the $O_2$ gas after a dehydration and dehydroxylation under a flow of $N_2$ (Chaplgin et al., 2010). The purified $O_2$ gas was analysed by dual-inlet isotope ratio mass spectrometry (IRMS, ThermoQuest Finnigan Delta Plus) against a working $O_2$ standard calibrated against VSMOW. Each analysis consisted in two runs of eight dual inlet measurements with an integration time of 26s. The sample isotope compositions were corrected on a daily basis using a quartz laboratory standard (Boulangé). During the measurement period,
Boulangé reproducibility (SD) was ± 0.13 ‰, ± 0.07 ‰ and ± 11 per meg for $\delta^{18}O$, $\delta^{17}O$ and $^{17}O$-excess respectively (n = 9). For a given sample two phytoliths aliquots were analyzed.

### 4.2. Leaf water extraction and analysis

Leaf water was extracted during 6 hours using a distillation line. Then a fluorination line was used to convert water to oxygen using $CoF_3$. Oxygen was analyzed by dual inlet IRMS (ThermoQuest Finnigan MAT 253)
against a working $O_2$ standard calibrated against VSMOW. The detailed procedure was previously described in Landais et al. (2006) and Alexandre et al. (2018). The reproducibility (2 replicates) was 0.015 ‰ for $\delta^{17}O$, 0.010 ‰ for $\delta^{18}O$ and 5 per meg for $^{17}O$-excess.

### 4.3. Irrigation and vaporization water analysis

The irrigation and vaporization waters were analyzed with an isotope laser analyzer (Picarro L2140i) operated
in $^{17}O$-excess mode using an auto-sampler and a high precision vaporizer as described in detail in Alexandre et al. (2018). The reproducibility (3 replicates) was 0.02 ‰, 0.01 ‰ and 10 per meg for $\delta^{17}O$, $\delta^{18}O$ and $^{17}O$-excess.

### 5. Results

#### 5.1. Phytolith concentrations, assemblages and origins in grass leaf (experiment 1)

In adult leaves, the phytolith concentration that ranges from 0.7 to 0.8% dry weight (d.w.) in sheaths and proximal blades increases to 2.1% d.w. in apical blades (Table 1a, fig. 2a). Thus, close to 60% of leaf phytoliths precipitate in apical blades. From young to adult and senescent blade, the phytolith content increases sharply from 0.8 to 1.4 and 3.0 % d.w. This means that close to 60% of blade phytoliths precipitate in the late period of leaf development, at the transition between adult and senescent stages.

Short cell phytoliths (fig. 1) are found in all samples but their relative abundance decreases from young to adult and senescent leaf blade and in adult leaf from sheath to proximal and apical blade (table. 1b). Conversely, long cell phytolith (fig.1) abundance increases with phytolith concentration from young to adult and senescent leaf blade (fig. 2a). Long cell phytoliths are absent from the sheath but reach 14% in proximal blade and 48% in apical blade. Parallelepipedal bulliform and Acicular hair phytoliths can be observed but in
small amount (<2%) in young leaves and senescent leaf blade samples. All phytolith assemblages contain thin silica particles with low refractive index, difficult to count with accuracy in light microscopy. SEM observation shows they are composed of multi-cellular silica sheets (mostly silicified cell walls and a few silicified stomata complexes) (fig. 1). Their abundance ranges from 16 to 33% in young and adult leaf blade and increases up to 52% in senescent leaf blades (Table 1b). In summary, silicification occurs mainly in the epidermis, starting
with short cells and continuing with long cells. Cell walls also silicify (silica sheets), in low proportion when the leaf is young or adult, in higher proportion when the leaf is close to senescence. Because these silica sheet



particles are very thin, their weight contribution to the isotope signature of bulk phytolith assemblages is expected to be significantly lower than their number.

### 5.2. Heterogeneity in the triple oxygen isotope composition of leaf water

Irrigation and leaf water (IW and LW, respectively) $\delta'^{18}O$, $\delta'^{17}O$, and $^{17}O$-excess values obtained from experiment 1 are presented in Table 1a and figure 2b. As expected, the lowest $\delta'^{18}O$ and $\delta'^{17}O$ values occur in the adult leaf sheath which is supposed to transpire less than the leaf blade (Webb and Longstaffe, 2003, 2000). The values of $\delta'^{18}O$ and $\delta'^{17}O$ are however higher than the irrigation water by 7.1 and 3.7 ‰ respectively, showing that transpiration still occurs. The $^{17}O$-excess of the leaf sheath water ($^{17}O$-excess$_{LW}$) is only slightly

lower (by 11 per meg) than $^{17}O$-excess$_{IW}$. In adult leaf waters, a clear evaporative fractionation trend occurs from the sheath to the proximal and apical blade. It is expressed by a decrease in $^{17}O$-excess$_{LW}$ (or $^{17}O$-excess$_{LW-IW}$) linearly correlated with an increase in $\Delta'^{18}O_{LW}$ (or $\Delta'^{18}O_{LW-IW}$) ($r^2= 0.99$, fig. 2b and 2c). Water from the young leaf blade plots on this line, close to water from the adult upper leaf blade (fig. 2b).

### 5.3. Heterogeneity in the triple oxygen isotope composition of leaf silica and leaf silica forming water

Polymerization of silica is supposed to occur in isotope equilibrium with the formation water, and, therefore, its isotope composition should only be governed by temperature and the isotope composition of the formation water (FW) (Alexandre et al., 2018; Dodd and Sharp, 2010; Sharp et al., 2016). The $\delta^{18}O$ of this formation water can be estimated from the temperature-dependent equation established by Dodd and Sharp (2010) from measurement of lake diatoms and lake water. At 20.4°C, the fractionation ($\Delta'^{18}O$) between water and silica

accounts for 31.07‰. Then, equation 10 from Sharp et al. (2016) can be used to calculate $\theta_{silica-water}$ ($\theta_{silica-water}$ = 0.5242) for the same temperature. Finally, the enrichment in $^{18}O$ and the difference in $^{17}O$-excess between phytolith forming water and irrigation water (respectively $\Delta'^{18}O_{FW-IW}$ and $^{17}O$-excess$_{FW-IW}$) can be estimated (Table 1a).

When plotted in the $^{17}O$-excess $vs$ $\Delta'^{18}O$ space (fig. 2c), the phytolith forming waters from adult leaf show a

clear evaporative fractionation trend from the sheath to the proximal and apical blade, similarly to the bulk leaf water. However, as early as in the sheath the phytolith forming water appears more impacted by evaporation than the bulk water. This discrepancy increases from the sheath to the proximal and apical blade (fig. 2c). This is also illustrated by the $\lambda_{FW-IW}$ values (ranging from 0.516 to 0.517) that are significantly lower than the $\lambda_{LW-IW}$ values (ranging from 0. 524 to 0.526) (table 1a). Variations in $\lambda_{FW-IW}$ or $\lambda_{LW-IW}$ values are too

small to be interpreted. Interestingly, the discrepancy between $^{17}O$-excess$_{LW-IW}$ and $^{17}O$-excess$_{FW-IW}$ is the same for the apical blade as for the bulk leaf (-119 and -129 per meg, respectively), in link with the dominance of apical blade phytoliths in the leaf phytolith assemblages (Table 1a).

The young leaf blades and senescence leaf blades phytolith forming waters plot on or close to the line in fig. 2c, and show similar $^{17}O$-excess$_{Phyto-IW}$ and $\Delta'^{18}O_{Phyto-IW}$ values than phytolith forming water in the adult leaf

blade, given the measurement precision.

In adult leaves, silicified short cell and silica sheets abundances decrease when $^{17}O$-excess$_{Phyto-IW}$ or $^{17}O$-excess$_{FW-IW}$ decrease from sheath to the proximal and apical leaf blade ($r^2> 0.8$). The inverse occurs for silicified long cells (fig. 2d, $r^2= 0.9$). As previously noted, young, adult and senescent leaf blade show similar $^{17}O$-excess$_{FW-IW}$ although the senescent leaf blade is characterized by an amount of silica sheets (52% of

counted particles) much higher than in young and adult leaf blades (24 and 18% of counted particles, respectively).

### 5.4. Effect of light/dark alternation on the triple oxygen isotope composition of leaf water and leaf silica.

Plant water isotope data from experiment 2a where light/dark alternations were set without changing RH, are

presented in Table 2. Variations within a given set of samples (e.g. F4-02-03-17 Day or F4-02-03-17 Night in Table 2) are important, alerting that interpretation in term of kinetic $vs$ equilibrium fractionation of small variations of $\Delta'^{18}O_{LW-IW}$ (<1‰), $^{17}O$-excess$_{LW-IW}$ (<14 per meg) or $\lambda_{LW-IW}$ (<0.0005) should be avoided. When





considering the margins of error, the averaged values of $\Delta'^{18}O_{LW-IW}$, $^{17}O\text{-excess}_{LW-IW}$ and $\lambda_{LW-IW}$ obtained after the dark period are similar to the ones obtained after the light period, suggesting that the contribution of
evaporated water to the bulk leaf water stayed constant under constant RH. It was not possible to measure the night and day transpiration flows or the stomatal conductance during the experiment.

In experiment 2b, transpiration and leaf blade phytolith concentrations are close when day/night alternation or constant day conditions are set (Table 2a). Phytolith concentrations range between 2.2 and 3.1% d.w, i.e. slightly higher than the adult leaf blade concentration obtained from experiment 1 (1.4% d.w.). A previous
experiment showed that silicification increases with the duration of growth and transpiration (Alexandre et al., 2018). In experiment 2b, the period of growth is shorter than in experiment 1 (16 $vs$ 39 days) but RH is lower during daytime (60 $vs$ 73%). Isotope data from experiment 2b (Table 2) shows that $\Delta'^{18}O_{Phyto-IW}$ and $^{17}O\text{-excess}_{Phyto-IW}$ values obtained for the same tank are very close when constant day conditions of light, temperature and RH or when day/night alternation conditions are set. $\lambda_{Phyto-IW}$ ranges from 0.520 to 0.521.

## 6. Discussion

### 6.1. Despite leaf water heterogeneity bulk leaf water triple oxygen isotope composition is predictable

In the conditions set for experiment 1, evaporative fractionation appears effective as soon as the water taken up by the roots reaches the sheath. This was expected as $^{18}O$-enrichment of several per mil relatively to the soil water was previously measured in the water of grass sheaths (Webb and Longstaffe, 2003). Indeed, stomata
are few but still present in the sheath (e.g. Chaffey, 1985) allowing weak transpiration (Martre et al., 2001). Here, the low amplitude of the $^{17}O\text{-excess}_{LW-IW}$ value (-11 per meg) supports that evaporated water weakly contributes to the water circulating in the sheath.

The strong evaporative $^{18}O$-enrichment observed from experiment 1 in the adult leaf water from the proximal to the apical blade (fig. 2b and 2c) was also expected. This has been commonly observed in monocot leaves
and was modelled using Péclet numbers (synthesis in Cernusak et al., 2016; Farquhar and Gan, 2003; Helliker and Ehleringer, 2000; Ogée et al., 2007). Briefly, after stomatal evaporation (evaporation model of Craig and Gordon, 1965), two main hydraulic processes combine in the leaf. The first one is a radial back-diffusion of evaporated $^{18}O$-enriched water from the epidermis to the mesophyll and into the veins, which leads to the second one, i.e. the progressive enrichment of the vein water from the base to the tip of the leaf, such as in the
case of a string of lakes (Gat and Bowser, 1991). In grasses, leaf blade is long and tapering, which influence interveinal distances (decreasing from bottom to tip). While stomatal densities and the capacity of water movement within the xylem remain relatively constant along the blade (Martre et al., 2001), the diffusional path length from vascular bundle to stomata decreases. Consequently, the stomatal conductance increases from bottom to tip (Ocheltree et al., 2012; Affek et al., 2006). This likely triggers increasing radial back-diffusion
of epidermal evaporated water which adds to the stem water already increasingly impacted by evaporation. The linear relationship obtained when plotting $^{17}O\text{-excess}_{LW-IW}$ $vs$ $\Delta'^{18}O_{LW-IW}$ for sheath, proximal and apical adult leaf blades of $F.\ arundinacea$ (fig. 2c) agrees with this pattern. In the $^{17}O\text{-excess}_{LW-IW}$ $vs$ $\Delta'^{18}O_{LW-IW}$ space, water of young blades plots close to water of the apical adult leaf blades as both blades are narrow with small diffusional path length to stomata favoring back-diffusion of evaporated water.

As an exercise, we estimated $\Delta'^{18}O_{LW-IW}$, $\Delta'^{17}O_{LW-IW}$, $^{17}O\text{-excess}_{LW-IW}$ and $\lambda_{LW-IW}$ for the bulk leaf blade water from experiment 1 (Table S1). For that purpose we used the Craig and Gordon model from Farquhar et al. (2007) (spreadsheet provided in Cernusak et al., 2016). The $^{17}O\text{-excess}_{LW-IW}$ and $\lambda_{LW-IW}$ values were calculated using equilibrium and kinetic fractionation factors (respectively $^{17}\alpha_{eq}$ and $^{17}\alpha_k$) of 0.529 and 0.518. We assumed that the grass transpiration had reached a steady state as climatic conditions were set constant during
the 39 days of growth. We additionally assumed that irrigation water and water vapour had the same isotope composition since i) there is no soil evaporation, ii) transpiration should not fractionate (e.g. Welp et al., 2008) and iii) the vaporised water comes from the same source as the irrigation water and is not fractionated by the vaporiser. Last, we measured the temperature of adult leaf of $F.\ arundinacea$ grown under conditions similar to those of experiment 1 (Table S2). No significant temperature difference was detected between the sheath,





lower and upper leaf blade because the leaf temperature was probably more controlled by light incidence than by evaporation. However, the leaf was systematically 2°C cooler than the surrounding air. In the model, when the leaf blade temperature was set similar to the air temperature, the predicted and observed values of $\Delta'^{18}O_{LW\text{-}IW}$, $\Delta'^{17}O_{LW\text{-}IW}$, $^{17}O\text{-excess}_{LW\text{-}IW}$ and $\lambda_{LW\text{-}IW}$ were significantly different (predicted $\Delta'^{18}O_{LW\text{-}IW}$ being higher than measured $\Delta'^{18}O_{LW\text{-}IW}$) (Table S1). When the observed temperature difference was included in the calculation,

the predicted values of $\Delta'^{18}O_{LW\text{-}IW}$, $\Delta'^{17}O_{LW\text{-}IW}$, $^{17}O\text{-excess}_{LW\text{-}IW}$ and $\lambda_{LW\text{-}IW}$ were similar to the observed values (Table S1). This result supports that under the experimental conditions of high RH, despite heterogeneity in the triple oxygen isotope composition of the water along the narrow blade, the bulk leaf blade water presents an averaged isotope signature that can be predicted by the single-water-source Craig and Gordon model. This result is also a reminder that, in addition to RH and the isotope signature of the source water, the temperature

difference between leaf and air is an important parameter to consider.

### 6.2. Bulk leaf water *vs* phytolith forming water triple oxygen isotope compositions

Experiment 1 shows that the epidermal water from which silica polymerizes is more impacted by evaporative isotope fractionation than the bulk water. This discrepancy occurs as early as in the sheath and increases from proximal to apical blade. This suggests that from bottom to tip the concentration gradient of evaporated water

that occurs radially in the leaf increases faster than the vertical concentration gradient in the veins. The oxygen isotope composition of evaporated water is very dependent on the concentration and isotope composition of the atmospheric water vapor around the leaf. Thus, experimentations including vapor triple oxygen isotope measurements are necessary to further assess which model accurately describes changes in the composition of epidermal water on the one hand and in bulk leaf water on the other hand.

### 6.3. Silicification patterns

The phytolith content and assemblages obtained from experiment 1 can be discussed in light of previous studies investigating silica deposition in grasses.

In a growing grass leaf, the cells are produced at the base of the leaf and are pushed towards the tip through the elongation zone during the growth (Kavanová et al., 2006; Skinner and Nelson, 1995). Silica precipitate

mainly in the epidermis. At the cell level, silicification, which is a rapid process taking a few hours (Kumar and Elbaum, 2017), initiates either in the extra-membranous space or in the cell wall and proceeds centripetally until the cell lumen is filled up (Bauer et al., 2011). During cell lumen silicification, some cells are still viable and transfer their content to each other before their full silicification (Kumar and Elbaum, 2017). Cell wall silicification, not followed by cell lumen full filling, has been also frequently observed, both in the epidermis

(Kumar et al., 2017) and in bundle sheath parenchyma cells surrounding the veins (Motomura, 2004). Over the course of leaf development, short cells are the first to silicify. Then, when the leaves become mature, long cell silicification takes over (Motomura, 2004; Kumar et al., 2016). Results from experiment 1, i.e. increase of long cell and decrease of short cell abundance from young to adult and senescent leaf blade, confirm this pattern. Experiment 1 additionally show that cell wall silicification (not followed by cell lumen filling) occurs

at all development stages and increases when the leaf reaches or is close to senescence.

Silicification processes can be metabolically controlled such as in the epidermal short cells, or passive, i.e. depending mainly on silica saturation during cell dehydration when the leaf transpires, such as in the epidermal long cells, bulliform cells, or hair cells (Kumar et al., 2017 and references therein, Kumar et al., 2019). The decrease in silicified short cell and silica sheet abundance from sheath to proximal and apical leaves when $^{17}O$-

$\text{excess}_{Phyto\text{-}IW}$ or $^{17}O\text{-excess}_{FW\text{-}IW}$ decreases (i.e. when evaporation increases) corroborates that evaporation (and induced transpiration) is not the main driver of these types of silicification. To the contrary, the strong negative relationship obtained between silicified long cells abundance and $^{17}O\text{-excess}_{Phyto\text{-}IW}$ confirms a strong control of evaporation on long cell silicification. This result has implications for paleoclimate reconstruction: in fossil phytolith assemblages, provided that the phytolith type relative abundances are not biased by taphonomical

processes, increasing long cell *vs* short cell phytoliths should covary with decreasing $^{17}O\text{-excess}_{Phyto}$ and increasing evaporative conditions.



### 6.4. Highlights for interpreting the triple oxygen isotope signature of grass phytolith assemblages

In nature, the biomass of grass sheath, stem, lower and upper blade, with different $^{17}$O-excess$_{Phyto}$ signatures, is highly species-dependant. However, if a large amount of silica polymerizes in the apical blades (52% of adult leaf silica in the case of *F. arundinacea*, Table 1), one may expect the RH dependant- evaporative fractionation impacting the apical blades to imprint the $^{17}$O-excess$_{Phyto}$ of all grass adult leaves.

Leaf senescence is a stress-induced or age-related developmental aging during which transpiration decreases to minimal level but is still efficient (Norton et al., 2014), epidermal conductance progressively prevailing over stomatal conductance (Smith et al., 2006). If the cells already contain dissolved silica, evaporation, not balanced by water input due to decreasing transpiration, may lead to silica saturation. Such a process likely occurs at the transition between the end of the elongation stage and the beginning of the senescence stage. Thus, if more than half of the silica, and of the silicified long cells polymerizes at that time (tables 1a and 1b), the isotope composition of the bulk silica sample is expected to be impacted by the soil water isotope composition and the climate conditions (RH, isotope composition of the water vapour, leaf *vs* atmosphere temperature difference) occurring during this transition. In experiment 1, climate conditions are constant when senescence occurs. In nature, seasonal climate change such as a drastic decrease of RH in the tropical and Mediterranean areas, are responsible for senescence, and may imprint the averaged isotope signature of grass phytoliths. Mere experimental monitoring of stress-induced (including changes in RH) senescence effect on $^{17}$O-excess$_{Phyto}$ could determine whether RH averaged over both elongation and senescence time lapses should be considered when interpreting $^{17}$O-excess$_{Phyto}$.

The results obtained from experiment 2a showed very close isotope composition of bulk leaf water during light and dark periods. The constancy of atmospheric relative humidity and temperature, as well as the shortness of experiment 2a (4 day/night light alternations after more than 2 months of constant day light) may have played against the closure of the stomata at night. A previous study on elongating leaves of *F. arundinacea* showed that spatial distribution of water content within the elongation zone can stay almost constant during the dark and light period (Schnyder and Nelson, 1988), supporting that dark/light alternations do not always impact the stomata openness. Assuming from experiment 2a that light/dark alternation has no impact on stomatal conductance, experiment 2b suggests that in case of RH variation from day to night, silica precipitation mainly occur when RH is the lowest (60% during the day instead of 80% during the night in experiment 2a), i.e. when evaporation and thus silica saturation are the highest. In the natural environment, nights are often characterized by lower temperature and higher RH than days, which should favor higher evaporation and silicification rates during the day. In these conditions, the triple oxygen isotope composition of grass leaf phytoliths should correlate more closely to daytime RH conditions than to day and night averaged RH conditions.

### 7. Conclusion

Increasing contribution of evaporated epidermal water to the bulk leaf water, from sheath to proximal and apical leaf blade, that has been previously observed in grass leaves, is well recorded by the triple oxygen isotope composition of *F. arundinacea* leaf water. Despite this heterogeneity, in the given experimental conditions (high RH), the bulk leaf blade water presents an averaged isotope signature that can be predicted by the single-water-source Craig and Gordon model.

Regarding silica, its forming water is mainly epidermal and is thus more impacted by evaporation than the bulk leaf water. This translates into lower $\lambda_{FW-IW}$ than $\lambda_{LW-IW}$ values. This discrepancy increases from sheath to proximal and apical blade and is likely driven by the steepening of the radial concentration gradient of evaporated water along the leaf. However, results from experiment 1 show that most of silica polymerizes in epidermal long cells of the apical blade. Thus, the triple oxygen isotope composition of apical blade phytoliths should imprint the isotope signal of bulk grass leaf phytoliths in nature, whatever is the diversity in grass anatomy.



Experiment 1 additionally show that most of silica polymerizes at the end of the elongation stage, close to senescence. Thus, climate conditions (RH, isotope composition of the water vapour, leaf *vs* atmosphere temperature difference) and soil water isotope composition occurring at the end of the elongation stage and at the transition towards senescence should be considered when interpreting $^{17}$O-excess$_{Phyto.}$ In nature, RH conditions leading to generalized leaf senescence should be considered, in addition to RH conditions prevailing during the grass leaf elongation, when interpreting $^{17}$O-excess$_{Phyto.}$.

No light/dark effect was detected on *F. arundinacea* transpiration and $^{17}$O-excess$_{Phyto-IW}$ in the frame of experiment 2. However, when day/night alternations are characterized by changes in RH, the lowest RH conditions will likely favor higher stomatal conductance and more evaporation leading to silica polymerization.

Both experiments contribute to a more precise identification of the parameters to take into consideration when interpreting the $^{17}$O-excess$_{Phyto.}$ They also brings elements to further understand and model the $\delta^{18}$O and $\delta^{17}$O of grass leaf water, which influences the isotope signal of several processes at the soil/plant/atmosphere interface.

*Acknowledgements*

This study was supported by the French program INSU-LEFE and by the ANR (HUMI-17 project). It benefited from the CNRS human and technical resources allocated to the ECOTRONS Research Infrastructures as well as from the state allocation 'Investissements d'Avenir' ANR-11-INBS-0001.

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





**Table 1. Growth chamber experiment 1 : a)** Experimental set-up, phytolith content and triple oxygen isotope data obtained for phytoliths (Phyto) and leaf water (LW). Isotope fractionation ($\Delta^{18}O$, $\Delta^{17}O$, $^{17}O$-excess and $\lambda$) between leaf water and irrigation water (LW-IW), phytoliths and irrigation water (Phyto-IW), phytoliths and leaf water (Phyto-LW) and phytolith-forming water/irrigation water (FW-IW) using $\Delta^{18}O_{silica-water}$ calculated after Dodd and Sharp (2010) and $\theta_{silica-water}$ calculated after Sharp et al. (2016, eq. 10), are presented. For irrigation water (IW), $\delta^{18}O_{IW}$ and $\delta^{17}O_{IW}$ are -5.59‰, -and -2.92‰, respectively. **b)** Results of phytolith counting.

Samples are named according to the climate chamber # they were collected in (P4), the set relative humidity (75%), the date of sampling (dd/mm/yy) and their anatomical origin. "Adult leaves: bulk blade av." is the weighted average of values obtained for proximal and apical blade samples.

Av : average ; n : number of replicates ; SD : standard deviation calculated on the replicates; Phyto Conc. (% d.w.) and Phyto proportion (%) stands for phytolith concentration expressed in % of the dry weight and phytolith proportion in % of adult leaf phytoliths. A mass loss correction factor of 0.7 was applied to the phytolith concentration in senescent leaves (see text for explanation).

a)

| Sample | Growing period | Dur. day | Temperature °C | SD | RH % | SD | Irrigation | Total biomass g | Phyto Conc. % | Phyto proportion % | | Phytoliths (Phyto) | | | | | | | | Leaf water (LW) | | | | | | | | LW-IW | | | | FW-IW | | | | Phyto-LW | | | | Phyto-IW | | | |
|---|---|---|---|---|---|---|---|---|---|---|---|---|---|---|---|---|---|---|---|---|---|---|---|---|---|---|---|---|---|---|---|---|---|---|---|---|---|---|---|---|---|---|---|---|
| | | | | | | | | | | | n | $\delta^{18}O$ ‰ | SD | $\delta^{17}O$ ‰ | SD | $\delta^{18}O$ ‰ | SD | $\delta^{17}O$ ‰ | SD | $^{17}O$-excess per meg | n | $\delta^{18}O$ ‰ | SD | $\delta^{17}O$ ‰ | SD | $\delta^{18}O$ ‰ | SD | $^{17}O$-excess per meg | $\Delta^{18}O$ ‰ | $\Delta^{17}O$ ‰ | $^{17}O$-excess LW-IW per meg | $\lambda$ | $\Delta^{18}O$ ‰ | $\Delta^{17}O$ ‰ | $^{17}O$-excess FW-IW per meg | $\lambda$ | $\Delta^{18}O$ ‰ | $\Delta^{17}O$ ‰ | $^{17}O$-excess Phyto-LW per meg | $\lambda$ | $\Delta^{18}O$ ‰ | $\Delta^{17}O$ ‰ | $^{17}O$-excess Phyto-IW per meg | $\lambda$ |
| **Experiment 1** | | | | | | | | | | | | | | | | | | | | | | | | | | | | | | | | | | | | | | | | | | | | |
| P4-75-02-09-2016-Y Young leaves: blade | 7/25 to 9/2/2016 | 39 | 20 | 0.1 | 73 | 0.9 | No | 57 | 0.8 | | 3 | 39.22 | 0.17 | 20.27 | 0.17 | 38.47 | 0.21 | 20.07 | 0.21 | -243 | 1 | 13.8 | | 7.20 | | 13.67 | 7.17 | -46 | 19.25 | 10.09 | -72 | 0.524 | 12.99 | 6.71 | -150 | 0.516 | 24.81 | 12.90 | -197 | 0.520 | 44.06 | 23.00 | -268 | 0.522 |
| P4-75-02-09-2016-5:h Adult leaves: sheath | 7/25 to 9/2/2016 | 39 | 20 | 0.1 | 73 | 0.9 | No | 56 | 0.8 | 30 | 3 | 35.32 | 0.09 | 18.30 | 0.12 | 34.71 | 0.09 | 18.13 | 0.12 | -193 | 1 | 1.54 | | 0.83 | | 1.54 | 0.83 | 15 | 7.12 | 3.75 | -11 | 0.526 | 9.22 | 4.77 | -100 | 0.522 | 33.17 | 17.31 | -208 | 0.522 | 40.29 | 21.06 | -219 | 0.523 |
| P4-75-02-09-2016-A-<10 Adult leaves: proximal blade | 7/25 to 9/2/2016 | 39 | 20 | 0.1 | 73 | 0.9 | No | 36 | 0.7 | 17 | 3 | 36.24 | 0.12 | 18.77 | 0.14 | 35.60 | 0.12 | 18.59 | 0.14 | -202 | 1 | 5.48 | | 2.89 | | 5.47 | 2.89 | 0.8 | 11.05 | 5.81 | -25 | 0.526 | 10.11 | 5.23 | -109 | 0.517 | 30.13 | 15.71 | -202 | 0.521 | 41.18 | 21.52 | -228 | 0.522 |
| P4-75-02-09-2016-A->10 Adult leave: apical blade | 7/25 to 9/2/2016 | 39 | 20 | 0.1 | 73 | 0.9 | No | 36 | 2.1 | 52 | 2 | 41.60 | 0.02 | 21.47 | 0.02 | 40.76 | 0.02 | 21.25 | 0.02 | -275 | 1 | 12.7 | | 6.63 | | 12.58 | 6.60 | -38 | 18.17 | 9.53 | -64 | 0.524 | 15.28 | 7.88 | -183 | 0.516 | 28.18 | 14.64 | -237 | 0.520 | 46.35 | 24.17 | -301 | 0.522 |
| Adult leaves: bulk blade av. | | | | | | | | 71 | 1.4 | | | 40.26 | | 20.80 | | 39.47 | | 20.58 | | -257 | | 9.09 | | 4.77 | | 9.04 | 4.76 | -19 | 14.63 | 7.68 | -45 | 0.525 | 13.99 | 7.22 | -164 | 0.516 | 28.67 | 14.91 | -228 | 0.520 | 45.06 | 23.51 | -282 | 0.522 |
| P4-75-02-09-2016-S Senescent leaves: blade | 7/25 to 9/2/2016 | 39 | 20 | 0.1 | 73 | 0.9 | No | 13 | 3.0 | | 3 | 39.66 | 0.19 | 20.51 | 0.23 | 38.89 | 0.23 | 20.30 | 0.23 | -235 | 7 | | | | | | | | | | | | | | | 0.517 | | | | | 44.48 | 23.22 | -261 | 0.522 |

b)

| Phytolith types | | Short cell (<6μm) | Short cell (>6μm) | Acicular hair | Bulliform cell | Silica sheets | Total | Long cells /(Long+Short cells) | Long cells /(Long+Silica sheets) |
|---|---|---|---|---|---|---|---|---|---|
| | | | | | | | | % | % |
| | % total | | | | | | | | |
| Young leaves: blade | | 52 | 22 | 0 | 1 | 24 | 724 | 29 | 47 |
| Adult leaves: sheath | | 67 | 0 | 0 | 0 | 33 | 643 | 1 | 1 |
| Adult leaves: proximal blade | | 61 | 14 | 0 | 0 | 24 | 534.3 | 19 | 37 |
| Adult leave: apical blade | | 34 | 48 | 2 | 1 | 16 | 385 | 59 | 76 |
| Adult leaves: bulk blade av. | | 40 | 40 | 2 | 0 | 18 | 509.8 | 49 | 69 |
| Senescent leaves: blade | | 15 | 30 | 2 | 1 | 52 | 423 | 67 | 36 |





**Table 2. Growth chamber experiment 2a and 2b.** Experimental set-up, phytolith content and triple oxygen isotope data obtained for phytoliths (Phyto), leaf water (LW) and irrigation water (IW). Isotope fractionation ($\Delta'^{18}O$, $\Delta'^{17}O$, $^{17}O$-excess and $\lambda$) between leaf water and irrigation water (LW-IW) and phytoliths and irrigation water (Phyto-IW) are presented.

Samples are named according to the climate chamber # they were collected in (e.g. F4), the date of sampling (dd/mm/yy) and the sampling after day or night (Day vs Night in experiment 2a) or after constant climate conditions or day/night alternation (Cst vs DN in experiment 2b).

n : number of replicates ; SD : standard deviation calculated on the replicates ; Phyto Conc. (% d.w.) stands for phytolith concentration expressed in % of the dry weight.

| Sample | | Growing p. | Dur. (day) | Temp. (°C) | RH (%) | Light (mmol/m2/sec) | Irrigation | Total biomass (g) | Transpiration (L/day) | Phyto Conc. (% d.w.) | n | $\delta^{18}O$ (‰) | SD | $\delta^{17}O$ (‰) | SD | $\delta'^{18}O$ (‰) | SD | $\delta'^{17}O$ (‰) | SD | $^{17}O$-excess (per meg) | SD | $\Delta'^{18}O$ (‰) | SD | $\Delta'^{17}O$ (‰) | SD | $^{17}O$-excess (per meg) | SD | $\lambda$ | SD |
|---|---|---|---|---|---|---|---|---|---|---|---|---|---|---|---|---|---|---|---|---|---|---|---|---|---|---|---|---|---|
| **Experiment 2a** | | | | | | | | | | | | Leaf water (LW) | | | | | | | | | | Leaf water - irrigation water (LW-IW) | | | | | | | |
| F4-02-03-17 Day-sample1 | Plant water | 12-16-2016 to 03-02-2017 | 22 | 25 | 60 | Sampled end 12h day | | | | | 1 | 6.80 | | 3.53 | | 6.78 | | 3.52 | | -58 | | 12.42 | | 6.47 | | -85 | | 0.521 | |
| F4-02-03-17 Day-sample2 | Plant water | 12-16-2016 to 03-02-2017 | 22 | 25 | 60 | Sampled end 12h day | | | | | 1 | 6.66 | | 3.45 | | 6.64 | | 3.44 | | -66 | | 12.28 | | 6.39 | | -92 | | 0.520 | |
| F4-02-03-17 Day-sample3 | Plant water | 12-16-2016 to 03-02-2017 | 22 | 25 | 60 | Sampled end 12h day | | | | | 1 | 6.23 | | 3.23 | | 6.21 | | 3.22 | | -55 | | 11.85 | | 6.18 | | -82 | | 0.521 | |
| F4-02-03-17 Day-sample4 | Plant water | 12-16-2016 to 03-02-2017 | 22 | 25 | 60 | Sampled end 12h day | | | | | 1 | 5.79 | | 3.00 | | 5.78 | | 2.99 | | -57 | | 11.42 | | 5.95 | | -84 | | 0.521 | |
| Average / SD | | | | | | | | | | | | 6.56 | 0.46 | 3.30 | 0.24 | 6.35 | 0.45 | 3.29 | 0.24 | -59 | 5 | 12.00 | 0.45 | 6.25 | 0.24 | -86 | 4.58 | 0.521 | 0.0003 |
| F4-06-03-17 Night sample5 | Plant water | 12-16-2016 to 03-02-2017 | 26 | 25 | 60 | Sampled end 12h night | | | | | 1 | 7.43 | | 3.85 | | 7.40 | | 3.84 | | -66 | | 13.05 | | 6.80 | | -92 | | 0.521 | |
| F4-06-03-17 Night sample6 | Plant water | 12-16-2016 to 03-02-2017 | 26 | 25 | 60 | Sampled end 12h night | | | | | 1 | 6.84 | | 3.54 | | 6.81 | | 3.54 | | -62 | | 12.46 | | 6.49 | | -89 | | 0.521 | |
| F4-06-03-17 Night sample7 | Plant water | 12-16-2016 to 03-02-2017 | 26 | 25 | 60 | Sampled end 12h night | | | | | 1 | 6.26 | | 3.25 | | 6.24 | | 3.25 | | -51 | | 11.89 | | 6.20 | | -77 | | 0.522 | |
| F4-06-03-17 Night sample8 | Plant water | 12-16-2016 to 03-02-2017 | 26 | 25 | 60 | Sampled end 12h night | | | | | 1 | 8.63 | | 4.46 | | 8.59 | | 4.45 | | -84 | | 14.24 | | 7.41 | | -111 | | 0.520 | |
| Average / SD | | | | | | | | | | | | 7.29 | 1.01 | 3.78 | 0.52 | 7.26 | 1.00 | 3.77 | 0.52 | -66 | 14 | 12.91 | 1.00 | 6.72 | 0.52 | -92 | 13.93 | 0.521 | 0.0005 |
| **Experiment 2b** | | | | | | | | | | | | Phytolith (Phyto) | | | | | | | | | | Phytolith - irrigation water (Phyto-IW) | | | | | | | |
| R2-16-12-2016-Cst | Phytolith | 11-30-2016 to 12-16-2016 | 16 | 25 | 60 | 290 | yes | 75 | | 2.2 | 1 | 33.80 | | 17.46 | | 33.24 | | 17.31 | | -239 | | 38.88 | | 20.27 | | -266 | | 0.521 | |
| R1-16-12-2016-DN | Phytolith | 11-30-2016 to 12-16-2016 | 16 | 25/15 | 60/80 | 290/0 alternation | yes | 83 | | 2.2 | 1 | 32.57 | | 16.82 | | 32.05 | | 16.68 | | -239 | | 37.69 | | 19.64 | | -266 | | 0.521 | |
| R2-03-01-2016-Cst | Phytolith | 12-16-2016 to 01-03-2017 | 18 | 25 | 60 | 290 | yes | 49 | 0.5 | 2.9 | 2 | 33.13 | 0.02 | 17.08 | 0.01 | 32.59 | 0.02 | 16.93 | 0.01 | -276 | 22 | 38.24 | | 19.89 | | -303 | | 0.520 | |
| R1-03-01-2016-DN | Phytolith | 12-16-2016 to 1-03-2017 | 18 | 25/15 | 60/80 | 290/0 alternation | yes | 48 | 0.4 | 3.1 | 2 | 34.50 | 0.03 | 17.76 | 0.02 | 33.92 | 0.03 | 17.60 | 0.02 | -304 | 4 | 39.56 | | 20.56 | | -330 | | 0.520 | |
| | | | | | | | | | | | | Irrigation water (IW) | | | | | | | | | | | | | | | | | |
| M2-16-12-16-Cst | irrigation water | 11-30 to 12-16-2016 | 16 | 25 | 60 | 290 | | | | | 3 | -5.63 | 0.011 | -2.95 | 0.023 | -5.65 | | -2.95 | | 27 | | | | | | | | | |
| M1-16-12-16 DN | irrigation water | 11-30 to 12-16-2016 | 16 | 25/15 | 60/80 | 290/0 alternation | | | | | 3 | -5.61 | 0.022 | -2.94 | 0.022 | -5.63 | | -2.94 | | 26 | | | | | | | | | |



**Figure 1.** Growth chamber experiment 1 : SEM pictures of phytoliths from young, adult and senescent leaf blades : silicifed Trapeziform short cell (1, 2 and 3), silicified Trapeziform sinuate short cell (4), undefined silicifed short cell or broken Elongate cylindric long cell (5), silicifed Elongate cylindric long cell (6a, 6b, 7 and 8), silicifed cell wall also reported as silica sheets (9 and 10).

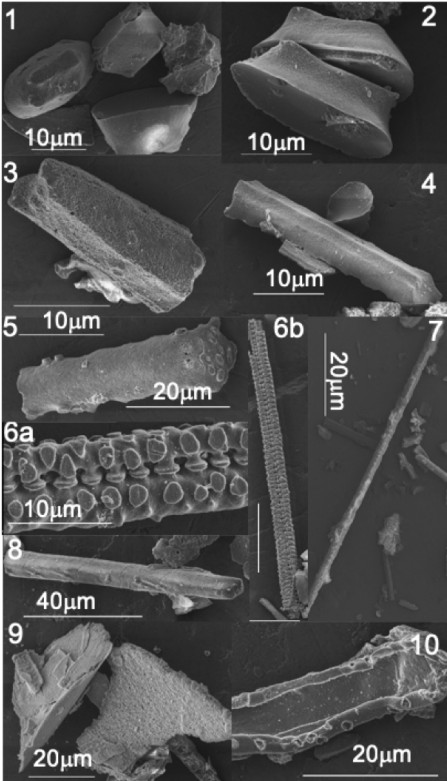



**Figure 2.** Growth chamber experiment 1 : **a)** Phytolith concentration *vs* silicified long cell proportion. Error bars represent the 5% error on counting (refer to text for details); **b)** Phytolith concentration *vs* silica sheet proportion. **c)** $^{17}O$-excess$_{LW-IW}$ and $^{17}O$-excess$_{FW-IW}$ vs $\Delta'^{18}O_{LW-IW}$ and $\Delta'^{18}O_{FW-IW}$; **c)** $^{17}O$-excess$_{FW-IW}$ vs silicified long cell, silicified short cell and silica sheet proportions. LW: leaf water, FW: phytolith forming water, IW: irrigation water. Correlation lines are drawn between adult sheath, blade>10cm and blade>10cm samples.

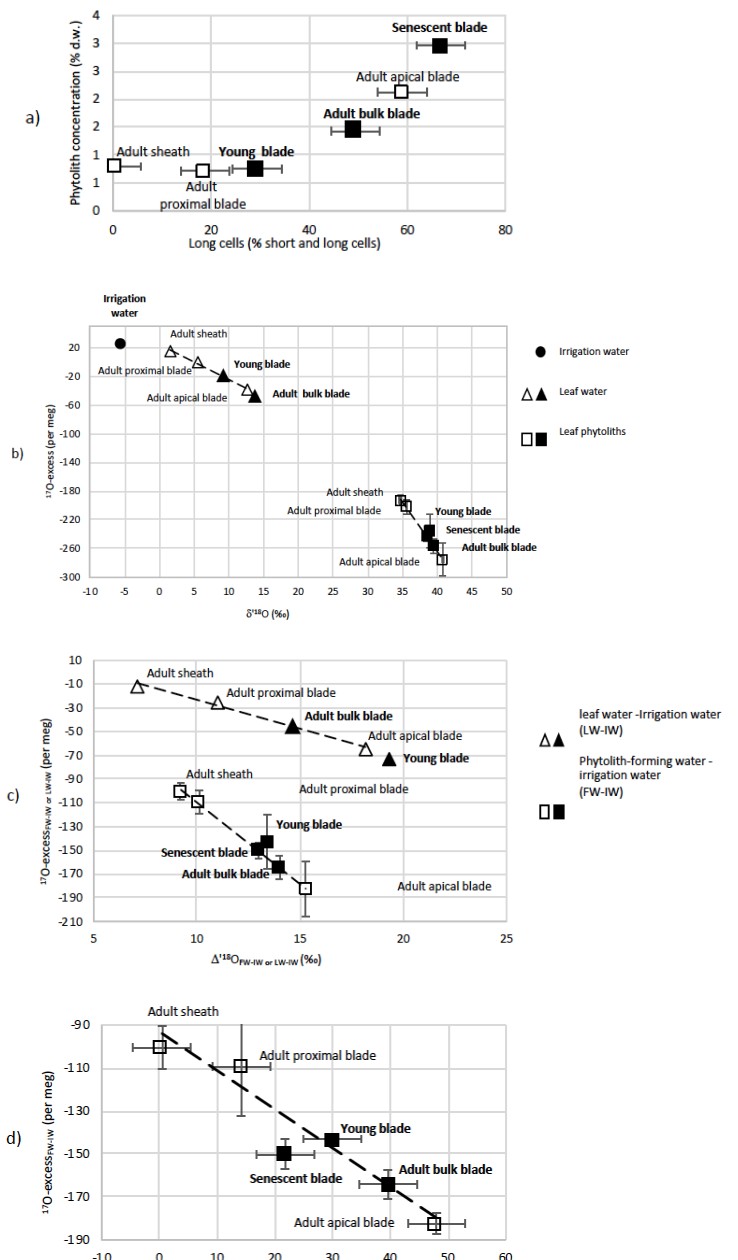