# Peer review of "Effects of leaf length and development stage on the triple oxygen isotope signature of grass leaf water and phytoliths: insights for a proxy of continental atmospheric humidity"

_Biogeosciences, 2019_

## Referee Comment (RC1) · Anonymous Referee #1 · 24 Apr 2019

The reviewed manuscript presents an important work, which examines the robustness of the 17O-excess of phytoliths as a proxy for RH. However, more work is needed to make the manuscript more readable. At the present form, I had troubles even verifying that the main conclusion are supported by the data. Moreover, I am not sure if there are quantitative conclusions, as vague terms like "most" are used throughout the text. Some detailed comments appear below.
Tables 1 and 2, contains full details of the results, and as such are hard to follow. So additional figures and summary tables are needed for easier access to the data (and then the full data can be moved to an appendix). Figure 2 should be divided into four different figures, each with its own caption. Will be good to explain in the caption what the filled versus empty markers stand for (this is also not explained in the legend). Figure 2a, correct the Y-axis.

The second half of the abstract needs to be made clearer. Start with: Lines 29 and 31: "most" – give percentage. Line 32: "at that time" – replace with "during this growth phase" for clarity Line 33 – Replace "At least", by "when RH was fixed during day/night" " However, when day/night alternations are characterized by significant changes in RH, the lowest RH conditions favoring evaporation and silica polymerization should be considered when calibrating the phytolith proxy." – Not clear. Do you mean the proxy is for the minimum diurnal RH? Also maybe need to add here what is the percentage of silica polymerization that happens under the lowest RH conditions.

The last sentence of the abstract should be removed or rewritten.

Section 2: The authors give there two different definition for theta. Since they are not identical, one of them must be wrong. Indeed the second one is the definition for lambda (or slope) which they give later. Papers cited by the authors give these definitions: 1) Angert, A., Cappa, C.D., DePaolo, D.J.: Kinetic O-17 effects in the hydrologic cycle: Indirect evidence and implications. Geochim. Cosmochim. Acta 68, 3487–3495, 2004. 2) Luz, B., Barkan, E.: Variations of 17O/16O and 18O/16O in meteoric waters. Geochim. Cosmochim. Acta 74, 6276–6286. https://doi.org/10.1016/j.gca.2010.08.016, 2010. Please correct. Also in Line 117, a reference line is defined not only by slope, but also by a point it is going through – add this info.

Line 287 – " Isotope data from experiment 2b (Table 2) shows that D'18OPhyto-IW and 17OexcessPhyto-IW values obtained for the same tank are very close when constant

day conditions of light,temperature and RH or when day/night alternation conditions are set." – This seems to contradict what appear in the abstract? So which one is correct? Do day/night variations in RH have an effect or not? Also, throughout the results, please don't just use the terms "very close" without giving the numbers, or indicating if the difference is significant or not.

Line 374 – If it is only "52%", why this will dominate the signal? Line 399 – "mainly occur" – give a number. Line 414 Quantify "most" and point to a figure or reference showing this. Line 426 – "Likely Favor"? – Still not clear to me. So what is the 17O-excess of phytoliths actually tell us? If the bottom line of this study is that more research is needed to answer this question, then this should be clearly written in the abstract and conclusions.

---

## Author Comment (AC1) · 3 Jul 2019

We thank Reviewer 1 for pointing out the weaknesses of the manuscript. Comments are answered and a revised draft is presented.

Please also note the supplement to this comment:
https://www.biogeosciences-discuss.net/bg-2019-73/bg-2019-73-AC1-supplement.zip

---

## Referee Comment (RC2) · Daniel Herwartz (Referee) · 23 Jul 2019

General comment: This paper examines the triple oxygen isotope systematics (expressed as 17Oexcess) of leaf water and the respective phytoliths. The overall goal is to use phytolith oxygen isotopic composition as a humidity proxy. A general correlation between phytolith 17Oexcess and relative humidity had already been established in a previous publication. The growth chamber experiments presented is this new paper aim to better understand the full spectrum of processes that may affect phyotolith 17Oexcess (e.g. dark/light cycles, changing T and RH with dark/light cycles). These experiments and the respective data are important to develop an accurate relative humidity proxy. I think the modelling part can be improved, which would also make this paper more significant for a broader scientific community.

Specific comments: In their experiment 1, the authors investigate how leaf water composition changes between different growth stages and along the leaf. In the revised version the authors modeled the expected evaporation trend by extending the model of Farquhar and Gan (2003) for 17Oexcess. The model curves are concave in triple oxygen isotope space and clearly differ from the convex model curves recently published for a series of evaporitic ponds (see Surma et al. 2018). Both models are based on the Craig and Gordon model, so this discrepancy comes unexpected. I discussed this puzzling observation with my PhD student Claudia Voigt and she discovered, that calculating the parameter h' both from $17\alpha$'s and $18\alpha$'s (not just $18\alpha$'s) changes the curvature of the model to a convex form (i.e. identical to Surma et al. 2018). At first sight it appears as if the data fit better to such a revised model but possibly more processes need to be considered (e.g. mixing). If the authors manage to improve the plant water model, their paper would become significant for many other fields. Leaf water controls the triple oxygen isotopic composition of O2 (produced from leaf water) and CO2 (equilibrates with leaf water). It's well worth the effort to improve the model.

I found it especially interesting, that the measured phytolith data cannot be modeled from measured leaf water using published equilibrium fractionation factors. If the published equilibrium fractionation factors are correct, kinetic effects must be responsible for the observed offset. Or the measured leaf water is not representative of local leaf water from which the phytoliths form. To me, the changing $\lambda$ values along the leaf seem to imply that the kinetic effects are not identical over the length of the blade. Is it possible to explain the data via contrasting fractionation factors during active (via enzymes) and/or passive (via evaporation) phytolith formation?

Line 120: The isotopic composition of the vapor in air is identical to that of irrigation water. If these two reservoirs have any chance to exchange, vapor in air would be driven to lower values (i.e. the two reservoirs equilibrate). The agar agar prevents such an exchange to some degree. I assume that water vapor in the air is constantly exchanged to ensure constant RH and vapor isotopic composition. Is this correct? The vapor isotopic composition has a strong effect on the evaporation trajectories in triple oxygen isotope space, so if partial equilibration occurs that would be important to know.

Line 231: The main reason why the sheath comprises a lower oxygen isotopic composition than the blade is not the lower transpiration rate. As a thought experiment, assume that transpiration rates in the sheath and the blade are identical. The 'source water' of the sheath would be irrigation water with low d18O. But the source water to the blade would be evaporated water from the sheath with somewhat enriched d18O. In this simple model the blade could have a far lower transpiration rate than the sheath and still comprise higher d18O.

Model for the prediction of phytoliths in Figure 3: The empirical $\lambda$Phyto-LW as calculated from this data is used to predict the triple oxygen isotopic composition of the photoliths, which is circular. If the published $\theta$silica-water = 0.524 is used, the 17Oexcess values would be far off (as shown in Figure S1). Present the model using $\lambda$=0.524 in Figure 1 (not only in Figure S1).

Section 7 (Conclusions): The first paragraph is confusing to me. Grass height and leaf height are mentioned here for the first time. Of course experiment 1 shows that leaf water composition changes along the leaf as predicted by the model, but this fractionation is not related to absolute hight but to l/lm. So a large (or high) leaf would carry the same bulk isotopic information as a short leaf (as stated at the end of paragraph 2). Also, I would not mix up the kinetic effects story with the RH story in the same paragraph.

Technical corrections: Line 57: Do not use the term distillation processes. In one of

the references you cite (Steig et al. 2014) a distillation experiment is conducted where 17Oexcess changes over 90per meg. Distillation processes can be governed both by equilibrium fractionation or kinetic fractionation depending on the set up. Line 124: provide 1 significant digit for the d18O isotopic composition. Line 190: Please specify how the working O2 gas was calibrated relative to SMOW or point to Alexandre et al. 2018. Provide the SMOW calibrated values for the internal quartz laboratory standard (Boulangé) and explain how that calibration was done. Ideally, provide a comparison of this laboratory internal standard to international standards with published D17O on SMOW scale. This is crucial for recalculating the data in case of any revised calibration. Line 215: Do you mean Figure 2 (not 1)? Line 230: Table 1? Line 235: The good fit of the linear correlation seems impressive at first sight but the irrigation water is not included in that regression. If the linear regression (presented in the first manuscript version) is extrapolated, the irrigation water clearly falls below the line. I advise against using linear regressions because evaporation trends are best represented by curves. Line 241: These $\lambda$Phyto-LW are significantly lower than the expected equilibrium fractionation between silicates and water ($\theta$silica-water = 0.524 for the 5-35°C temperature range). The average reader won't remember that value so you may want to note that discrepancy here. Line 252: Remind the reader that RH and T changed with the light/dark alternations in this experiment. Line 287: The second ii) should be iii). Line 304: source not tsource. Table S3: If the leaf temperature is reduced from 20.4 to 18.4, the RH at the site of evaporation changes, so RH with respect to the leaf temperature (not air temperature) should be used as also recommended by Farquhar and Gan (2003). The Reference list is missing in the revised version. Caption of Fig. 3: $17\alpha$ = $18\alpha\lambda$ not $17\alpha$ = $17\alpha\lambda$ Clean up the legend of Fig. 3. (e.g. use $\lambda$=0.52x)

---

## Author Comment (AC2) · 6 Sep 2019

We are very grateful to Daniel Herwartz, assisted by Claudia Voigt, for his in-depth review. As advised, we further worked the modelling part. All the issues raised by the reviewer will be addressed in a revised draft. Answers, point by point, are following in supplementary material.

[Figure]

Please also note the supplement to this comment:
https://www.biogeosciences-discuss.net/bg-2019-73/bg-2019-73-AC2-supplement.zip
* * *

---

## Author Response (AR1)

We thank D. Herwartz, C.Voigt and an anonymous reviewer for their in depth reviews that substantially improved the modelling approach. All their comments were considered in the present revised draft, as listed below. All modifications that were made after the submission of the first revised draft (in the course of the BG-discussion) are in blue in the present revised draft. An author (Clément Outrequin, PhD student at CEREGE) was added to the list of authors as discussions with Clément greatly helped to clarify the modelling approach in the discussion section of the revised draft.

**Reviewer 1**

*The reviewed manuscript presents an important work, which examines the robustness of the 17O-excess of phytoliths as a proxy for RH. However, more work is needed to make the manuscript more readable. At the present form, I had troubles even verifying that the main conclusion are supported by the data. Moreover, I am not sure if there are quantitative conclusions, as vague terms like "most" are used throughout the text.*

The text was rewritten to make the objectives, results, interpretation, conclusions and abstract clearer. Vague terms were avoided.

- The two questions to be dealt with are more clearly stated in the introduction: 1) whether grass anatomy diversity impacts the $^{17}$O-excess of phytoliths *vs* RH relationship , 2) whether RH changes from day to night and from leaf elongation to leaf senescence should be considered when interpreting the the $^{17}$O-excess of phytoliths as a RH proxy.
- Two section (6.2 and 6.3) now explain how we can model the triple oxygen isotope composition of leaf water and phytoliths along the blade. They show that evolution with length of $^{17}$O-excess of *F. arundinacea* leaf water and phytoliths can be predicted using the Farquhar and Gan (2003) model and considering a $\lambda_{silica\text{-}water}$ value decreasing from 0.522 to 0.520 from the sheath to the apical part of the blade. Despite of this heterogeneity, the $^{17}$O-excess and d'$^{18}$O values of bulk leaf water and phytoliths can be estimated and are not length-dependent. Impact of the diversity in grass physiognomy on the triple oxygen isotope composition of phytoliths is discussed in more details leading to the conclusion that it should not impact the triple oxygen isotope composition of bulk grass phytoliths.
- We now make clearer that as most of silica polymerizes at the end of the elongation stage (58 % in the present case), RH conditions leading to leaf senescence in nature should be considered in addition to RH condition during leaf elongation, when interpreting the $^{17}$O-excess of phytoliths.
- Tracks for future research are given when inconclusive results are acquired (i.e. potential kinetic fractionation during phytolith formation, impact of day/night alternation on the triple oxygen isotope composition of phytoliths).
- The abstract, discussion and conclusion are made clearer

*Tables 1 and 2, contains full details of the results, and as such are hard to follow. So additional figures and summary tables are needed for easier access to the data (and then the full data can be moved to an appendix).*

Tables 1 is simplified. Table 2 is moved to supplementary material. Two tables showing the modelled calculations for the triple oxygen isotope composition of leaf water and phytoliths along the blade are added (Tables S3 and S4).

*Figure 2 should be divided into four different figures, each with its own caption. Will be good to explain in the caption what the filled versus empty markers stand for (this is also not explained in the legend).*
*Figure 2a, correct the Y-axis.*

Figures were reworked for further clarity. Only 3 figures are now presented:

- **Figure 1.** Growth chamber experiment 1 : SEM pictures of phytoliths from young, adult and senescent leaf blead : silicifed Trapeziform short cell (1, 2 and 3), silicified Trapeziform sinuate short cell (4), undefined silicifed short cell or broken Elongate cylindric long cell (5), silicifed Elongate cylindric long cell (6a, 6b, 7 and 8), silicifed cell wall also reported as silica sheets (9 and 10).
- **Figure 2. Growth chamber experiment 1 : a)** Phytolith concentration *vs* Long Cell phytolith proportion. Error bars represent the 5% error on counting (refer to text for details).

- **Figure 3. Growth chamber experiment 1: leaf water and phytolith triple oxygen isotope data and estimates.** Observed $^{17}O$-excess vs $\delta'^{18}O$ for leaf water (triangles) and phytoliths (squares) in young, adult and senescent leaves (black symbols) and along adult leaf (sheath, proximal blade, apical blade) (white symbols) (Table 1). Error bars are displayed when smaller than the symbols. Estimated $^{17}O$-excess $vs$ $d'^{18}O$ for bulk leaf water (Table S3) and along the leaf length (Table S4) according to the Craig and Gordon (C&G) model (Cernusak et al., 2016; Farquhar et al., 2007), and the C&G model complemented with a mixing equation.

*The second half of the abstract needs to be made clearer. Start with: Lines 29 and 31: "most" – give percentage. Line 32: "at that time" – replace with "during this growth phase" for clarity Line 33 – Replace "At least", by "when RH was fixed during day/night"" However, when day/night alternations are characterized by significant changes in RH, the lowest RH conditions favoring evaporation and silica polymerization should be considered when calibrating the phytolith proxy." – Not clear. Do you mean the proxy is for the minimum diurnal RH? Also maybe need to add here what is the percentage of silica polymerization that happens under the lowest RH conditions. The last sentence of the abstract should be removed or rewritten.*

The abstract was rewritten for more clarity/accuracy:

Continental relative humidity (RH) is a key climate parameter but there is a lack of quantitative RH proxies suitable for climate model-data comparisons. Recently, a combination of climate chamber and natural transect calibrations laid the groundwork for examining the robustness of the triple oxygen isotope composition ($\delta'^{18}O$ and $^{17}O$-excess) of phytoliths, that preserve in sediments, as a new proxy for past changes in RH. However, it was recommended that besides RH, additional factors that may impact $\delta'^{18}O$ and $^{17}O$-excess of plant water and phytoliths be examined. Here, the effects of grass leaf length, leaf development stage and day/night alternations are addressed from growth chamber experiments. The triple oxygen isotope compositions of leaf water and phytoliths of the grass species *F. arundinacea* are analyzed. Evolution of the leaf water $\delta'^{18}O$ and $^{17}O$-excess along the leaf length can be modelled using a string of lake approach to which an unevaporated-evaporated mixing equation must be added. We show that for phytoliths to record this evolution, a kinetic fractionation between leaf water and silica, increasing from the base to the apex, must be assumed. Despite the isotope heterogeneity of leaf water along the leaf length, the bulk leaf phytolith $\delta'^{18}O$ and $^{17}O$-excess values can be estimated from the Craig and Gordon model and a mean leaf water-phytolith fractionation exponent ($\lambda_{Phyto-LW}$) of 0.521. In addition to not being leaf length-dependent, $\delta'^{18}O$ and $^{17}O$-excess of grass phytoliths are expected to be impacted only very slightly by the stem $vs$ leaf biomass ratio. Our experiment additionally shows that because a lot of silica polymerizes in grasses when the leaf reaches senescence (58% of leaf phytoliths in mass), RH prevailing during the start of senescence should be considered in addition to RH prevailing during leaf growth when interpreting the $^{17}O$-excess of grass bulk phytoliths. Although under the study conditions $^{17}O$-excess$_{Phyto}$ do not vary significantly from constant day to day/night conditions, additional monitoring at low RH conditions should be done before drawing any generalizable conclusions. Overall, this study strengthens the reliability of the $^{17}O$-excess of phytoliths to be used as a proxy of RH. If future studies show that the mean value of 0.521 used for the grass leaf water-phytolith fractionation exponent $\lambda_{Phyto-LW}$ is not climate-dependent, then grassland leaf water $^{17}O$-excess obtained from grassland phytolith $^{17}O$-excess would inform on isotope signals of several soil-plant-atmosphere processes.

*Section 2: The authors give there two different definition for theta. Since they are not identical, one of them must be wrong. Indeed the second one is the definition for lambda (or slope) which they give later. Papers cited by the authors give these definitions: 1) Angert, A., Cappa, C.D., DePaolo, D.J.: Kinetic O-17 effects in the hydrologic cycle: Indirect evidence and implications. Geochim. Cosmochim. Acta 68, 3487–3495, 2004. 2) Luz, B., Barkan, E.: Variations of 17O/16O and 18O/16O in meteoric waters. Geochim. Cosmochim. Acta 74, 6276–6286. https://doi.org/10.1016/j.gca.2010.08.016, 2010. Please correct.*

The section was corrected as follows (section 2):

In the oxygen triple isotope system ($\delta^{18}O$, $\delta^{17}O$), the fractionation factors ($^{17}\alpha$ and $^{18}\alpha$ are related by the exponent $\theta$ where $^{17}\alpha = {}^{18}\alpha^{\theta}$ or $\theta = \ln^{17}\alpha / \ln^{18}\alpha$. For the silica-water couple and according to the Sharp et al. (2016) empirical equation 10, $\theta_{silica-water}$ equals 0. 524 for the 5-35°C temperature range. For the water liquid-water vapor couple at equilibrium, $\theta_{equil}$ equals 0.529 for the 11-41°C range (Barkan and Luz, 2005). When evaporation occurs, a fractionation due to the vapor diffusion in air is added to the equilibrium fractionation, as conceptualized by the Craig and Gordon model (Craig and Gordon, 1965; Gat, 1996). $\theta_{diff}$ associated with this diffusion fractionation equals 0.518 (Barkan and Luz, 2007). When RH decreases, amplitude of the fractionation governed by $\theta_{diff}$ increases. While $\theta$ applies to a particular well constrained physical process, the term $\lambda$ is used when several fractionation processes occur at the same time. The overall fractionation in the triple oxygen isotope system can be formulated as following: $\lambda = \Delta'^{17}O_{A-B} / \Delta'^{18}O_{A-B}$ with $\Delta'^{17}O_{A-B} = \delta'^{17}O_A - \delta'^{17}O_B$, $\Delta'^{18}O_{A-B} = \delta'^{18}O_A - \delta'^{18}O_B$, $\delta'^{17}O = \ln(\delta^{17}O + 1)$ and $\delta'^{18}O = \ln(\delta^{18}O + 1)$. $\delta$ and $\delta'$ notations are expressed in ‰ *vs* VSMOW. In the $\delta'^{18}O$ vs $\delta'^{17}O$ space, $\lambda$ represents the slope of the line linking $\Delta'^{17}O_{A-B}$ to $\Delta'^{18}O_{A-B}$.

***Also in Line 117, a reference line is defined not only by slope, but also by a point it is going through – add this info.***

Changed in section 2: In the $\delta'^{17}O$ *vs* $\delta'^{18}O$ space, the $^{17}O$-excess depicts the $\delta'^{17}O$ departure from a reference line with a slope $\lambda$ of 0.528. This is the slope of the Global Meteoric Water Line (expressed as $\delta'^{17}O = -0.528 \times \delta'^{18}O + 0.33$ per meg, Luz and Barkan, 2010).

***Line 287 – " Isotope data from experiment 2b (Table 2) shows that D'18OPhyto-IW and 17OexcessPhyto-IW values obtained for the same tank are very close when constant day conditions of light,temperature and RH or when day/night alternation conditions are set." – This seems to contradict what appear in the abstract? So which one is correct? Do day/night variations in RH have an effect or not? Also, throughout the results, please don't just use the terms "very close" without giving the numbers, or indicating if the difference is significant or not.***

Text and abstract were made clearer: in the experimental conditions, day/night alternations do not modify the triple oxygen isotope composition of leaf phytoliths. However, additional monitoring for low RH conditions should be performed before withdrawing any generalizable conclusions.

***Line 374 – If it is only "52%", why this will dominate the signal? Line 399 – "mainly occur" – give a number. Line 414 Quantify "most" and point to a figure or reference showing this. Line 426 – "Likely Favor"? – Still not clear to me. So what is the 17Oexcess of phytoliths actually tell us? If the bottom line of this study is that more research is needed to answer this question, then this should be clearly written in the abstract and conclusions.***

As previously stated, the text was rewritten to make the objectives, results, interpretation, conclusions and abstract clearer. Vague terms were avoided. Overall, the presented data and estimates contribute to a more precise identification of the parameters to take into consideration when using the $^{17}O$-excess$_{Phyto}$ *vs* RH relationship previously obtained (Alexandre et al., 2018).

**Reviewer 2**

***In their experiment 1, the authors investigate how leaf water composition changes between different growth stages and along the leaf. In the revised version the authors modeled the expected evaporation trend by extending the model of Farquhar and Gan (2003) for 17Oexcess. The model curves are concave in triple oxygen isotope space and clearly differ from the convex model curves recently published for a series of evaporitic ponds (see Surma et al. 2018). Both models are based on the Craig and Gordon model, so this discrepancy comes unexpected. I discussed this puzzling observation with my PhD student Claudia Voigt and she discovered, that calculating the parameter h' both from 17_'s and 18_'s (not just 18_'s) changes the curvature of the model to a convex form (i.e. identical to Surma et al. 2018).***

Thanks for pointing out this inaccuracy. We revised the model accordingly:

In Table S4, h is the ratio of ambient humidity to the humidity at the sites of evaporation in the leaf. The parameter h' (h'= 1-a$_{equil}$ * a$_k$ *(1-h)) (Farquhar and Gan, 2003) depends on h and the equilibrium and diffusion isotope fractionation values which differ for $^{18}O$ and $^{17}O$ (Table S3 of the excel file). As a consequence, h' calculated for $^{18}O$ ($^{18}$h' in Table S4) is different from h calculated for $^{17}O$ ($^{17}$h' in table S4). Table S4 is corrected accordingly and the revised model curves (Figure 3 of the revised draft presented below) are convex, as the ones presented in Surma et al. (2018) for evaporated lakes.

**Figure 3. Growth chamber experiment 1: leaf water and phytolith triple oxygen isotope data and estimates**

Observed $^{17}O$-excess vs '$^{18}O$ for leaf water (triangles) and phytoliths (squares) in young, adult and senescent leaves (black symbols) and along adult leaf (sheath, proximal blade, apical blade) (white symbols) (Table 1). Error bars are displayed when smaller than the symbols. Estimated $^{17}O$-excess *vs* '$^{18}O$ for bulk leaf water (Table S3) and along the leaf length (Table S4) according to the Craig and Gordon (C&G) model (Cernusak et al., 2016; Farquhar et al., 2007), and the C&G model complemented with a mixing equation.

[Figure]

*At first sight it appears as if the data fit better to such a revised model but possibly more processes need to be considered (e.g. mixing).If the authors manage to improve the plant water model, their paper would become significant for many other fields. Leaf water controls the triple oxygen isotopic composition of O2 (produced from leaf water) and CO2 (equilibrates with leaf water). It's well worth the effort to improve the model.*

In agreement with this comment a mixing equation is added to the Craig and Gordon model, as explained in the discussion section of the revised draft:

[revised manuscript text omitted]
$_{\text{Phyto}}$ and $^{17}$O-excess$_{\text{Phyto}}$ in grass leaves. $\lambda_{\text{Phyto-LW}}$ value being lower than the $\theta_{\text{silica-water}}$ value of 0.524 calculated after Sharp et al. (2016) implies that either the $\theta_{\text{silica-water}}$ value previously established is overestimated or a kinetic fractionation occurs during phytolith formation. Our modeling exercise suggests that the amplitude of such a kinetic fractionation would increase from the base to the apex of the leaf ($\lambda_{\text{Phyto-LW}}$ decreasing regularly from 0.522 to 0.520). The proportion of short cell phytoliths for which silica polymerization is genetically controlled, decreases from the base to the apex (Table S2). This would go against a kinetic fractionation enzymatically controlled. However, further knowledge on the mechanisms of silica polymerization is needed to further discuss this point.

Positions of the phytolith data on the modeled phytolith curve are not exactly the same as positions of the leaf water data on the modeled leaf water curve, especially for the apical part. This discrepency suggests that E in the apical leaf water is higher than E in the phytolith-forming water. This can be explained assuming the following: phytolith-forming water integrates the whole grass elongation period (Kumar and Elbaum, 2017; Kumar et al., 2016, 2019) while the sampled leaf water only represents a snapshot. In grass leaf, the epidermal cells close to the apex were produced at the base of the leaf and pushed upper during the growth. Hence apical epidermal cells are older than the cells close to the base and phytoliths in these cells gather early and late phytoliths formed at low and high l/lm values with low and high E values, respectively.

For the grass bulk leaf, despite this discrepancy along leaf length, and assuming a mean $\lambda_{\text{Phyto-LW}}$ of 0.521, $\delta'^{18}$O$_{\text{Phyto}}$ and $^{17}$O-excess$_{\text{Phyto}}$ record $\delta'^{18}$O$_{\text{LW}}$ and $^{17}$O-excess$_{\text{LW}}$. In other terms, whatever the grass leaf length, $\delta'^{18}$O$_{\text{Phyto}}$ and $^{17}$O-excess$_{\text{Phyto,}}$ should be determinable from the Craig and Gordon model complemented by an unevaporated-evaporated water mixing equation. The main controls on $\delta'^{18}$O$_{\text{Phyto}}$ and $^{17}$O-excess$_{\text{Phyto}}$ are thus the soil water and vapor isotope compositions, the difference of temperature between leaf water and atmosphere, RH, and E.

***I found it especially interesting, that the measured phytolith data cannot be modeled from measured leaf water using published equilibrium fractionation factors. If the published equilibrium fractionation factors are correct, kinetic effects must be responsible for the observed offset. Or the measured leaf water is not representative of local leaf water from which the phytoliths form. To me, the changing _ values along the leaf seem to imply that the kinetic effects are not identical over the length of the blade. Is it possible to explain the data via contrasting fractionation factors during active (via enzymes) and/or passive (via evaporation) phytolith formation?***
As presented above, this is now assessed in the discussion section 6.3. :

The comparison between modeled and observed isotope compositions brings insights on the factors driving $\delta'^{18}$O$_{\text{Phyto}}$ and $^{17}$O-excess$_{\text{Phyto}}$ in grass leaves. $\lambda_{\text{Phyto-LW}}$ value being lower than the $\theta_{\text{silica-water}}$ value of 0.524 calculated after Sharp et al. (2016) implies that either the $\theta_{\text{silica-water}}$ value previously established is overestimated or a kinetic fractionation occurs during phytolith formation. Our modeling exercise suggests that the amplitude of such a kinetic fractionation would increase from the base to the apex of the leaf ($\lambda_{\text{Phyto-LW}}$ decreasing regularly from 0.522 to 0.520). The proportion

of short cell phytoliths for which silica polymerization is genetically controlled, decreases from the base to the apex (Table S2). This would go against a kinetic fractionation enzymatically controlled. However, further knowledge on the mechanisms of silica polymerization is needed to further discuss this point.

*Line 120: The isotopic composition of the vapor in air is identical to that of irrigation water. If these two reservoirs have any chance to exchange, vapor in air would be driven to lower values (i.e. the two reservoirs equilibrate). The agar agar prevents such an exchange to some degree. I assume that water vapor in the air is constantly exchanged to ensure constant RH and vapor isotopic composition. Is this correct? The vapor isotopic composition has a strong effect on the evaporation trajectories in triple oxygen isotope space, so if partial equilibration occurs that would be important to know.*

As described in the Material section, ambient relative humidity is kept constant in the growth chamber by combining a flow of dry air and an ultrasonic humidifier that produces vapor without any isotope fractionation. Thus, yes, the water vapor in the air should be constantly exchanged at the growing chamber scale, although, we cannot completely rule out that right above the agar-agar and around the leaves some water vapor comes from the soil water evaporation. Additional (and in progress) experiment including continuous measurement of the atmospheric water vapor will help to further assess this point in a near future.

*Line 231: The main reason why the sheath comprises a lower oxygen isotopic composition than the blade is not the lower transpiration rate. As a thought experiment, assume that transpiration rates in the sheath and the blade are identical. The 'source water' of the sheath would be irrigation water with low d18O. But the source water to the blade would be evaporated water from the sheath with somewhat enriched d18O. In this simple model the blade could have a far lower transpiration rate than the sheath and still comprise higher d18O.*

Indeed, in this model the isotope composition along the leaf does not depend on transpiration. Thanks for highlighting this inaccuracy which is corrected in the revised draft. The revised model is now based on the only assumption that the initial water is the irrigation water. The model clearly shows the $^{18}$O-enrichment in the sheath water and successive enrichment in the blade (Figure 3 of the revised draft).

*Model for the prediction of phytoliths in Figure 3: The empirical _Phyto-LW as calculated from this data is used to predict the triple oxygen isotopic composition of the photoliths, which is circular. If the published _silica-water = 0.524 is used, the 17Oexcess values would be far off (as shown in Figure S1). Present the model using _=0.524 in Figure 1 (not only in Figure S1).*

Figure 3 of the revised draft now present the modeled isotope composition of phytoliths assuming $l_{Phto-LW}$ of 0.524, 0.522 and decreasing from 0.522 to 0.520.

*Section 7 (Conclusions): The first paragraph is confusing to me. Grass height and leaf height are mentioned here for the first time. Of course experiment 1 shows that leaf water composition changes along the leaf as predicted by the model, but this fractionation is not related to absolute hight but to l/lm. So a large (or high) leaf would carry the same bulk isotopic information as a short leaf (as stated at the end of paragraph 2). Also, I would not mix up the kinetic effects story with the RH story in the same paragraph.*

Thanks for underlying this inaccuracy. The conclusion was rewritten for further clarity as follows:

**7. Conclusions**

The data and estimates presented here contribute to a more precise identification of the parameters to take into consideration when using the $^{17}$O-excess$_{Phyto}$ as a RH proxy (Alexandre et al., 2018). Neither grass height nor grass physiognomy should significantly impact the isotope composition of bulk grass leaf water and phytoliths. By contrast, RH prevailing at the start of senescence should be considered in addition to RH prevailing during leaf growth when interpreting $^{17}$O-excess$_{Phyto}$. If future studies show that the fractionation between leaf water and phytoliths, expressed by a mean $\lambda_{Phyto-LW}$ value of 0.521, is not climate-dependent, then the triple oxygen isotope composition of bulk leaf water should be obtainable from the triple oxygen isotope composition of grassland phytolith assemblages. The parameters driving the triple oxygen isotope

composition of both grass leaf water and phytoliths are given by the Craig and Gordon model applied to leaves (Farquhar et al., 2007) and the unevaporated-evaporated water mixing equation. Thus the most important paramaeters are the difference between soil water and vapor isotope compositions, the difference between leaf and atmosphere temperatures, RH, and E. Being able to record the triple oxygen isotope composition of grassland leaf water would bring some significant insights into i) estimating the triple oxygen isotope composition of $CO_2$ equilibrated with leaf water and partitioning gross fluxes of $CO_2$ from vegetation at the regional scale (e.g. Helliker and Ehleringer, 2000) or ii) estimating at the global scale the triple oxygen isotope composition of $O_2$ produced by the biosphere and quantifying its productivity from air bubbles trapped in ice cores (Blunier et al., 2002).

**Technical corrections**

*Line 57: Do not use the term distillation processes. In one of the references you cite (Steig et al. 2014) a distillation experiment is conducted where 17Oexcess changes over 90per meg. Distillation processes can be governed both by equilibrium fractionation or kinetic fractionation depending on the set up.*
This is modified in the revised draft (introduction): The $\delta^{18}O$ and $\delta^{17}O$ combination varies weakly in precipitation (Angert et al., 2004; Barkan and Luz, 2007; Landais et al., 2008) and is not significantly affected by temperature (Barkan and Luz, 2005; Uemura et al., 2010), in contrast to the deuterium-excess (d-excess = $\delta^2H - 8.0 \times \delta^{18}O$).

*Line 124: provide 1 significant digit for the d18O isotopic composition.*
This is corrected in the revised draft: two digits are given for every $\delta^{18}O$ value

*Line 190: Please specify how the working O2 gas was calibrated relative to SMOW or point to Alexandre et al. 2018. Provide the SMOW calibrated values for the internal quartz laboratory standard (Boulangé) and explain how that calibration was done. Ideally, provide a comparison of this laboratory internal standard to international standards with published D17O on SMOW scale. This is crucial for recalculating the data in case of any revised calibration.*
The paragraph was reworked accordingly in section 4.1:
Phytoliths triple oxygen isotope analysis was performed as described in details in Alexandre et al. (2018). The IR Laser-Heating Fluorination Technique (Alexandre et al., 2006, Crespin et al., 2008; Suavet et al., 2010) was used to extract the oxygen gas ($O_2$) after dehydration and dehydroxylation under a flow of $N_2$ (Chapligin et al., 2010). Then, the $O_2$ was passed through a -114°C slush to refreeze gases interfering with the mass 33 (e.g. NF). These interfering gases may be produced during the fluorination of residual N in the line. The purified $O_2$ was sent to a dual-inlet mass spectrometer (ThermoQuest Finnigan Delta Plus). The composition of the reference gas was determined through the analyses of NBS28 for which isotope composition has been set to $\delta^{18}O$ = 9.60 ‰ *vs* VSMOW, $\delta^{17}O$ = 4.99‰ *vs* VSMOW and $^{17}O$-excess = 65 per meg. Each analysis consisted of two runs of eight dual inlet measurements with an integration time of 26 seconds. The sample isotope compositions were corrected on a daily basis using a quartz laboratory standard (Boulangé) with $\delta^{18}O$ = 16.284 ‰ *vs* VSMOW, $\delta^{17}O$ = 8.463 ‰ *vs* VSMOW. During the measurement period, Boulangé reproducibility (SD) was ± 0.13 ‰, ± 0.07 ‰ and ± 11 per meg for $\delta^{18}O$, $\delta^{17}O$ and $^{17}O$-excess respectively (n = 9). For a given sample, from two to three phytoliths aliquots were analyzed. Measured reproducibility ranged from 5 to 23 per meg.

*Line 215: Do you mean Figure 2 (not 1)?*
Corrected

*Line 230: Table 1?*
fig. 1, Table S2

*Line 235:The good fit of the linear correlation seems impressive at first sight but the irrigation water is not included in that regression. If the linear regression (presented in the first manuscript version) is extrapolated, the irrigation water clearly falls below the line. I advise against using linear regressions because evaporation trends are best represented by curves.*

Right. In the revised draft curves represent the evaporation trends

*Line 241: These _Phyto-LW are significantly lower than the expected equilibrium fractionation between silicates and water (_silica-water = 0.524 for the 5-35_C temperature range). The average reader won't remember that value so you may want to note that discrepancy here.*

Corrected

*Line 252: Remind the reader that RH and T changed with the light/dark alternations in this experiment.*

Corrected

*Line 287: The second ii) should be iii).*

Corrected

*Line 304: source not tsource.*

Corrected

*The Reference list is missing in the revised version.*

The reference list will be added in the revised draft.

*Caption of Fig. 3: 17_ = 18__ not 17_ = 17__*
*Clean up the legend of Fig. 3. (e.g. use _=0.52x)*

This is corrected in Figure 3 of the revised draft.